# THE SPATIAL BLINDSPOT OF VISION-LANGUAGE MODELS

## ABSTRACT

Vision-language models (VLMs) have advanced rapidly, but their ability to capture spatial relationships remains a critical blindspot. Current VLMs are typically built with contrastive language-image pretraining (CLIP) style image encoders. The training recipe often flattens images into 1D patch sequences, fundamentally discarding the 2D structure necessary for spatial reasoning. We argue that this lack of spatial awareness is a missing dimension in VLM design and a key bottleneck for applications requiring strong multimodal grounding, such as robotics and embodied AI. To address this, we investigate two overlooked components: (i) image encoders trained with alternative objectives and (ii) 2D positional encodings. Our work shows that these architectural choices lead to models with superior spatial reasoning, highlighting a key but underexplored design space for grounded AI. Code for this work will be released soon.

## 1 INTRODUCTION

Visual understanding in VLMs usually relies on pre-trained image encoders (Dosovitskiy et al., 2021; Radford et al., 2021b; Sun et al., 2023; Oquab et al., 2024) for image representation. These encoders can be frozen (Zhang et al., 2024; Tang et al., 2024; Alayrac et al., 2022; Liu et al., 2023c; Li et al., 2023) or partially unfrozen (Jeong et al., 2024). The reliance on pre-trained encoders can lead to poor performance in downstream tasks. Tong et al. (Tong et al., 2024b) observe that CLIP (Radford et al., 2021a) focuses on the overall semantic understanding, overlooking the details of the visual system. Anis et al. (Anis et al., 2025) find that state-of-the-art VLMs like CLIP (Radford et al., 2021a) and SigLIP (Zhai et al., 2023) can exhibit poor performance with simple image transformations such as rotation or flipping. This suggests that their understanding of spatial relationships might not be invariant to changes in viewpoint or orientation, which is a crucial aspect of spatial understanding. This limitation impedes their effectiveness in complex tasks that require intricate manipulation, precise navigation, and a deeper understanding of how objects relate to each other in both space and time. Tong et al. (Tong et al., 2024b) build a stronger image encoder by combining DINOv2 (Oquab et al., 2024) and CLIP that excels at both general semantic and dense image understanding. TIPS (Maninis et al., 2025) shows that combining the contrastive image-text style of training (Radford et al., 2021a) along with the self-supervised masked image modeling in DINOv2 style can help models better understand the spatial relationship in the visual world without any spatially aware image annotation in the training dataset. Recent advances in image encoders (Tschannen et al., 2025; Fini et al., 2024) explore alternative design and training strategies than CLIP, but their usage in the construction of VLMs is minimal.

That brings us to the next topic of how the image embeddings of these encoders are aligned with the LLM backbone for training VLMs. The prevalent approach flattens image tokens into a 1D sequence before applying RoPE-1D (Su et al., 2023). Recent work (Zhang et al., 2025a;b) sheds light on how RoPE-1D can lead to lack of spatial awareness in current VLMs. A notable exception here is Qwen2-VL (Wang et al., 2024) which introduces multimodal rotary positional embedding (M-RoPE). M-RoPE decomposes positional encoding into temporal, height, and width components; preserving the physical and temporal aspects of the visual information.

We also observe that foundational VLMs (Alayrac et al., 2022; Liu et al., 2023c; Li et al., 2023; Xiao et al., 2024) typically do not report their performance on spatial reasoning benchmarks, despite the fact that spatial understanding is a fundamental aspect of visual perception. Current evaluations

often emphasize general vision-language tasks such as VQA, captioning, or retrieval, but these fail to capture whether models can reason about relative positions, geometric relations, or fine-grained spatial layouts. Without standardized reporting on such capabilities, progress in spatial reasoning will remain limited.

We address the above challenges and make the following contributions:

1. We systematically evaluate several state-of-the-art VLMs on a set of spatial reasoning benchmarks, revealing significant performance gaps in their ability to model geometric relations and spatial layouts.

2. We investigate the impact of alternative image encoders, including SigLIP (Zhai et al., 2023), SigLIP2 (Tschannen et al., 2025), and AIMv2 (Fini et al., 2024) on the spatial reasoning capabilities of VLMs within the LLaVA framework. Our findings highlight the role of encoder choice in spatial understanding.

3. we apply 2D-RoPE in image encoder–LLM alignment process to encode both x and y co-ordinates of an image pixel. We demonstrate that this approach improves the preservation of 2D spatial information and enhances the spatial reasoning performance of VLMs.

## 2    RELATED WORK

### 2.1    IMAGE ENCODERS

In recent years, we have seen progress on image encoders aligned with their own text captioning. CLIP (Radford et al., 2021b) aligns image and text representations through a large-scale contrastive loss on paired dataset, and demonstrates strong zero-shot generalization across classification and retrieval tasks. In contrast, DINOv2 (Oquab et al., 2024) is a visual model trained with a self-supervised masked image modeling objective. This allows DINOv2 to learn rich, pixel-level features by predicting masked-out image patches, without relying on paired text supervision. BLIP (Li et al., 2022) proposed a unified framework that integrates both contrastive pre-training for vision and generative objectives such as image–text matching and captioning. BLIP demonstrated that combining discriminative and generative signals leads to richer cross-modal representations and improved downstream transfer. BLIP-2 (Li et al., 2023) further advanced this line by introducing a lightweight querying transformer that bridges frozen image encoders with large language models, enabling efficient vision–language alignment without end-to-end pretraining of massive multimodal models. SigLIP (Zhai et al., 2023) improves CLIP by replacing the softmax contrastive loss with a pairwise sigmoid loss, removing batch-size dependencies and resulting in more stable training dynamics and efficient scaling. SigLIP2 (Tschannen et al., 2025) extends this line of work by augmenting the SigLIP training recipe with self-distillation and masked prediction objectives, resulting in higher-quality dense features, and improved localization across vision–language benchmarks. SigLIP2 also introduces the NaFlex variant, which preserves native aspect ratios and supports variable sequence lengths. The NaFlex variant provides empirical gains on aspect-sensitive tasks such as OCR and document understanding. In contrast to these contrastive and bridging approaches, AIMv2 (Fini et al., 2024) introduces an autoregressive multimodal pretraining framework that unifies vision and language by pairing a vision encoder with a decoder trained to jointly generate image patches and text tokens. Cocchi et al. (Cocchi et al., 2025) conduct a study that systematically pairs various LLMs with different visual backbones such as CLIP, DINOv2, SigLIP, and SigLIP2 to understand the strengths and limitations of various VLM integration strategies. Our work differs from theirs as we focus on spatial awareness along with the integration of M-RoPE in image encoders; specifically 2D-RoPE for images.

### 2.2    SPATIAL REASONING

Zhang et al. (Zhang et al., 2025a) classified spatial reasoning into two categories, static vs. dynamic. Static spatial reasoning refers to understanding spatial relationships in fixed configurations, where objects do not change positions over time. These tasks test whether a model can identify, compare, or infer spatial arrangements from still images or single states of a scene. For example, determining whether 'the book is to the left of the laptop' or 'the chair is behind the table' requires recognizing and reasoning about stable relative positions. Static reasoning often emphasizes relational concepts

such as distance, orientation, containment, or perspective, without involving temporal changes. Dynamic spatial reasoning, on the other hand, involves scenarios in which the spatial configuration evolves. Here, models must account for motion, transformation, or sequential changes in a scene. Typical examples include predicting the outcome of an object's movement, tracking shifting perspectives, or reasoning about cause-and-effect actions in space. Unlike static reasoning, dynamic tasks demand temporal and often causal understanding, requiring models to build an implicit representation of change and transformation. Another way to categorize spatial reasoning is 2D vs. 3D images. 2D reasoning focuses on relationships within 2D images, such as relative position, alignment, or counting, whereas 3D reasoning requires understanding depth, perspective, and volumetric relations. In this paper, we focus on static spatial awareness in 2D images.

## 2.3 VLMs and Spatial Awareness

Recent advances in LLMs and general purpose image encoders, such as CLIP (Radford et al., 2021b) and SigLIP (Zhai et al., 2023) have significantly improved VLM capabilities. Architectures such as Flamingo (Alayrac et al., 2022), LLaVA (Liu et al., 2023c;b), KOSMOS (Pan et al., 2023; Peng et al., 2023), Florence-2 (Xiao et al., 2024), and Molmo (Deitke et al., 2024) demonstrate strong performance in tasks such as image captioning, visual question answering (VQA) and complex reasoning. Qwen2-VL (Wang et al., 2024) introduced multimodal rotary position encoding (M-RoPE) and dynamic resolution techniques, while PaLI's joint modality scaling and cross-lingual learning (Chen et al., 2022; 2023) have improved vision-language understanding. These VLMs tend to rely on implicit visual features learned during training rather than explicitly structured representations of spatial information needed for world understanding. RoboSpatial (Song et al., 2025) is an annotated dataset specifically designed for spatial understanding in VLMs that includes 1 million images, 5,000 3D scans, and 3 million annotated spatial relationships. The pairing of 2D egocentric images with 3D scans in RoboSpatial makes it suitable for both 2D and 3D spatial understanding tasks. SpatialVLM (Chen et al., 2024) focuses on training VLMs with spatial reasoning dataset on the internet to improve their 3D spatial understanding. This framework enhances the ability of VLMs to recognize quantitative relationships between physical objects, such as distances and size differences, which are crucial for many real-world tasks. SpatialVLM has also shown potential in robotics as a tool for providing fine-grained reward annotations. MM-Spatial (Daxberger et al., 2025) introduces CA-VQA, a new supervised fine-tuning dataset and benchmark focused on indoor scenes. It targets tasks such as spatial relationship reasoning, metric estimation, and 3D grounding. By training MM-Spatial on CA-VQA, the model achieves state-of-the-art performance in 3D spatial understanding, highlighting the value of incorporating depth and multiview cues. SpatialRGPT (Cheng et al., 2024) introduces a data curation pipeline that enables effective learning of regional representations from 3D scene graphs. It also features a flexible plugin module for integrating depth information into the visual encoder of existing VLMs. This allows SpatialRGPT to accurately perceive the relative directions and distances of user-specified regions within a visual scene.

## 2.4 Measuring Spatial Awareness

There are a number of benchmarks evaluating VLMs on spatial awareness; spanning across tasks such as 2D layout understanding, 3D geometric reasoning, counting, and relational question answering. Multimodal Visual Pattern (MMVP) benchmark (Tong et al., 2024b) is a set of 300 images in 9 categories, designed by observing CLIP-blind pairs. CV-Bench (Tong et al., 2024a) includes both the 2D and 3D tasks around spatial relationships, object counting, depth ordering, and relative distance estimation. GQA (Hudson & Manning, 2019) leverages scene graph annotations and functional programs to generate compositional questions, and includes subsets like compare, verify, and logical that specifically test spatial relations such as left, right, on, and under. Visual Spatial Reasoning (VSR) (Liu et al., 2023a) assesses fine-grained relational understanding through 66 annotated spatial relations in natural images, while TopViewRS (Li et al., 2024b) measures reasoning over semantic top-view maps. TallyQA (Acharya et al., 2019) and CountBenchQA (Beyer et al., 2024) address object numerosity under occlusion and clutter.

# 3 EXPERIMENTAL SETUP

## 3.1 METHODS

Our experiment is based on the LLaVA (Liu et al., 2023c) framework as shown in Figure 1. To explore spatial awareness in VLMs, we extend the LLaVA architecture by integrating alternative image encoders beyond CLIP along with 2D-RoPE to preserve the inherent 2D spatial structure as shown in Figure 2. Specifically, we experiment with CLIP, SigLIP, SigLIP2, AIMv2 along with a modified design to include 2D-RoPE. Our hypothesis is that these encoder designs will better capture hierarchical spatial cues and contextual relationships across image regions. Unlike standard RoPE, 2D-RoPE encodes both horizontal and vertical patch indices through concatenated sinusoidal embeddings, providing explicit spatial priors for attention. In our implementation, 2D-RoPE is applied after patch embeddings are projected into query and key vectors, ensuring that spatial relations are injected directly into the attention mechanism rather than at the raw patch level. We hypothesize that these design choices will build a VLM with better relative position understanding in 2D images.

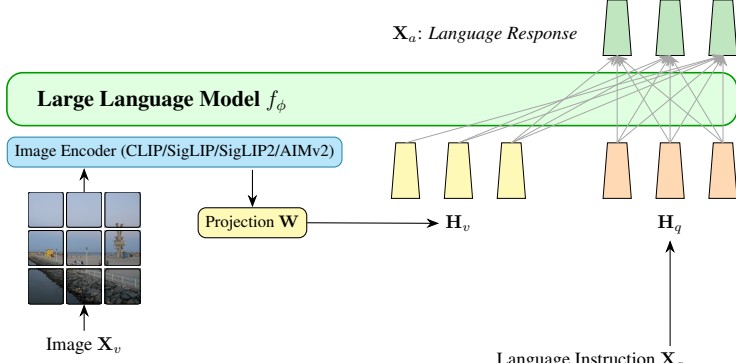

Figure 1: Our experimental approach with LLaVA Framework Liu et al. (2023c) that compares the performance of different image encoders and 2D-RoPE variants.

## 3.2 TRAINING STRATEGIES

We train our models in two stages: pretraining and instruction fine-tuning. The training process uses the same dataset used in LLaVA pretraining and instruction tuning respectively. For each image $X_v$, we used the single-turn conversation data

$$(X_q^1, X_a^1, \ldots, X_q^T, X_a^T)$$

where $T$ is the total number of turns from LLaVA.

We conduct our experiments using a cluster of 8 NVIDIA H100 GPUs, each with 80 GB of VRAM. For both pretraining and fine-tuning, we resize input images to a resolution of 256×256 pixels.

For pretraining, we employ a per-device batch size of 32, which results in a global batch size of 256. We optimize the model using the AdamW optimizer with a learning rate of 1e-3, and a cosine learning rate scheduler is applied. The pretraining process, which focuses solely on training the projection matrix, takes approximately 3 hours.

For fine-tuning, we perform a full model update. The fine-tuning configuration uses a learning rate of 2e-5, per-device batch size of 16, resulting in a global batch size of 128. This process takes approximately 24 hours for each experiment.

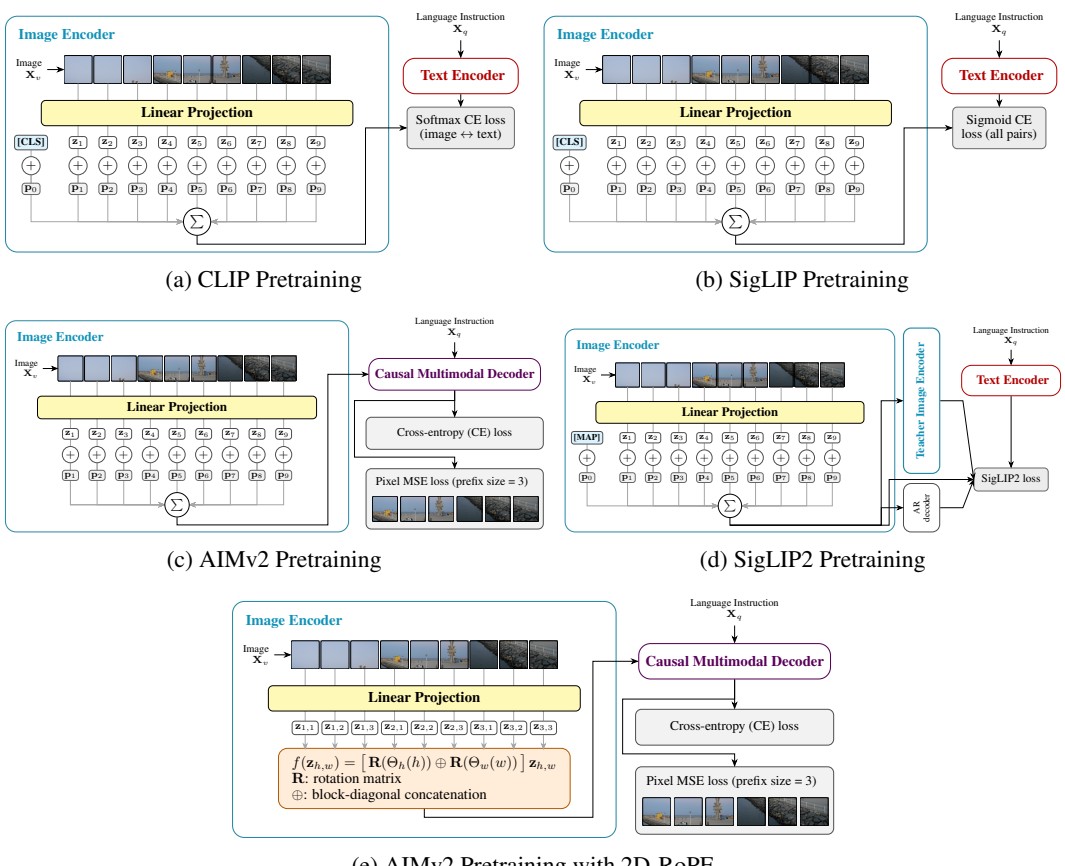

Figure 2: Pretraining strategies with various image backbones and 2D-RoPE

## 4 RESULTS

We compare frontier models and LLaVA variants on various benchmarks. Our work is based on the LLaVA-1.5 7B model. Therefore, all encoders and their 2D-RoPE variants that we trained are 7B models. Frontier models between the 2B to 8B parameter range were compared with the LLaVA variants we trained. For comparison purposes, we used LLaVA-NeXT 7B (Liu et al., 2024), LLaVA-OneVision-qwen2-7B-ov-hf (Li et al., 2024a), Qwen2.5-VL-8B (Bai et al., 2025), SmolVLM2-2.2B-Instruct (Marafioti et al., 2025), Gemma3-4b-it (Team et al., 2025), PaliGemma2-3b-mix-448 (Beyer et al., 2024) and Molmo-7B-D-0924 (Deitke et al., 2024).

### 4.1 EVALUATING ON GENERAL PURPOSE BENCHMARKS

As shown in Table 1, Qwen2.5-VL achieves the strongest overall performance, obtaining the highest scores on MMMU_Val, MME, CCBench, and SEEDBench-IMG. Among the LLaVA variants, AIMv2 demonstrates the most consistent results, particularly on MME and SEEDBench-IMG, although its 2D-RoPE counterpart provides only marginal gains on certain metrics and about 1.75% drop in MME. The effect of 2D-RoPE is mixed across variants. For example, LLaVA-SigLIP improves about 10% on MMMU_Val, while other models show little benefit or even reduced performance. LLaVA-SigLIP-2D-RoPE achieves the strongest MMMU_Val score among LLaVA encoders, but AIMv2 remains more competitive across multiple benchmarks.

### 4.2 EVALUATING ON SPATIAL BENCHMARKS

In Table 2, Qwen2.5-VL stands out as the strongest frontier model overall, achieving the highest performance on CV-Bench 2D, MMVP, VSR, TopViewRS, and CountBenchQA. LLaVA-NeXT leads

Table 1: Comparison of multimodal models across MMMU_Val, MME, CCBench, and SEEDBench-IMG. Values underlined indicate the best-performing model overall, while values in **bold** highlight the best-performing LLaVA encoder variant in each benchmark.

| Models | MMMU_Val | MME | CCBench | SEEDBench-IMG |
|---|---|---|---|---|
| Qwen2.5-VL | 0.580 | 0.826 | 0.592 | 0.770 |
| LLaVA-NeXT | 0.376 | 0.632 | 0.243 | 0.696 |
| LLaVA-OneVision | 0.479 | 0.712 | 0.549 | 0.767 |
| SmolVLM2 | 0.416 | 0.640 | 0.231 | 0.713 |
| Gemma3-4b-it | 0.473 | 0.620 | 0.369 | 0.655 |
| PaliGemma | 0.307 | 0.580 | 0.333 | 0.715 |
| Molmo | 0.491 | 0.662 | 0.367 | 0.746 |
| LLaVA-1.5 | 0.322 | 0.589 | 0.084 | 0.601 |
| LLaVA-2D-RoPE | 0.298 | 0.582 | **0.086** | 0.585 |
| LLaVA-SigLIP | 0.303 | 0.559 | 0.071 | 0.567 |
| LLaVA-SigLIP-2D-RoPE | **0.334** | 0.525 | 0.080 | 0.545 |
| LLaVA-SigLIP2 | 0.309 | 0.495 | **0.086** | 0.548 |
| LLaVA-SigLIP2-2D-RoPE | 0.323 | 0.542 | **0.086** | 0.527 |
| LLaVA-AIMv2 | 0.314 | **0.573** | **0.086** | **0.595** |
| LLaVA-AIMv2-2D-RoPE | 0.311 | 0.563 | 0.114 | 0.586 |

Table 2: Comparison of frontier models and LLaVA variants across spatial understanding benchmarks. Values underlined indicate the best-performing frontier model; values in **bold** indicate the best-performing LLaVA variant.

| Models | MMVP | CV-Bench 2D Overall | TallyQA | GQA Overall | VSR | Top-ViewRS | Count-BenchQA |
|---|---|---|---|---|---|---|---|
| LLaVA-NeXT | 0.667 | 0.606 | 0.733 | 63.786 | 63.994 | 0.409 | 0.515 |
| LLaVA-OneVision | 0.767 | 0.730 | 0.797 | 62.140 | 77.741 | 0.414 | 0.823 |
| Qwen2.5-VL | 0.770 | 0.754 | 0.800 | 60.391 | 89.116 | 0.456 | 0.891 |
| SmolVLM2 | 0.687 | 0.577 | 0.729 | 50.574 | 71.277 | 0.416 | 0.692 |
| Gemma3-4b-it | 0.708 | 0.659 | 0.525 | 31.277 | 55.074 | 0.334 | 0.713 |
| PaliGemma | 0.667 | 0.624 | 0.794 | 62.570 | 65.139 | 0.322 | 0.674 |
| Molmo | 0.753 | 0.728 | 0.808 | 55.295 | 76.432 | 0.323 | 0.858 |
| LLaVA v1.5 | 0.577 | 0.490 | 0.707 | 33.225 | 55.810 | 0.384 | 0.468 |
| LLaVA-2D-RoPE | 0.513 | 0.443 | 0.654 | 34.433 | 57.201 | 0.283 | 0.290 |
| LLaVA-SigLIP | 0.433 | 0.412 | 0.672 | 25.648 | 54.910 | 0.349 | 0.581 |
| LLaVA-SigLIP-2D-RoPE | 0.507 | 0.425 | 0.616 | **38.448** | 57.692 | 0.295 | 0.483 |
| LLaVA-SigLIP2 | 0.427 | 0.442 | 0.684 | 23.970 | 52.701 | **0.371** | 0.532 |
| LLaVA-SigLIP2-2D-RoPE | 0.480 | 0.415 | 0.646 | 34.560 | 56.465 | 0.330 | 0.402 |
| LLaVA-AIMv2 | 0.513 | **0.466** | **0.710** | 32.541 | 56.219 | 0.339 | **0.739** |
| LLaVA-AIMv2-2D-RoPE | **0.560** | 0.432 | 0.690 | 32.342 | **60.311** | 0.338 | 0.719 |

frontier models on GQA Overall, while Molmo achieves the best TallyQA score among frontiers, indicating complementary strengths across different tasks. Among the LLaVA variants, performance is more fragmented: LLaVA-AIMv2 shows the most consistent improvements, reaching the highest scores on CV-Bench 2D, TallyQA, and CountBenchQA. LLaVA-AIMv2-2D-RoPE improves MMVP and leads on VSR. LLaVA-SigLIP-2D-RoPE dominates GQA Overall, and LLaVA-SigLIP2 leads in TopViewRS. We observe that although the LLaVA variants surpass Gemma3-4b-it in TallyQA, VSR; their overall performance did not surpass other frontier models. We think this is because of how Gemma3-4b-it is designed. Gemma3-4b-it uses SigLIP variant as its vision encoder, but the fixed resolution of the encoder with Pan & Scan algorithm. Pan & Scan is a preprocessing algorithm that allows the model to handle high-resolution and non-square images by breaking them down into smaller, fixed-size crops. This is necessary because Gemma 3's vision encoder can only process images at a fixed resolution of 896x896 pixels. This design leads to information loss that hinders

Figure 3: Example image from LLaVA-Bench (In-the-Wild) (Liu et al., 2023c).

Table 3: Model outputs for the prompt *Are the chopsticks to the left or right of the bowl?* on image shown in Figure 3.

| Model | Output |
|---|---|
| LLaVA-v1.5 | The chopsticks are to the right of the bowl. |
| LLaVA-2D-RoPE | The chopsticks are to the right of the bowl. |
| LLaVA-SigLIP | The chopsticks are to the right of the bowl. |
| LLaVA-SigLIP-2D-RoPE | The chopsticks are to the right of the bowl. |
| LLaVA-SigLIP2 | The chopsticks are to the right of the bowl. |
| LLaVA-SigLIP2-2D-RoPE | Right |
| LLaVA-AIMv2 | The chopsticks are to the right of the bowl. |
| LLaVA-AIMv2-2D-RoPE | The chopsticks are to the right of the bowl. |
| Qwen2.5-VL | The chopsticks are to the right of the bowl. |
| **Gemma3-4b-it** | **The chopsticks are to the left of the bowl.** |

performance in benchmarks that require fine-grained visual reasoning, such as VSR and TallyQA. In Table 3, we observe that Gemma3-4b-it mistakenly thinks that the chopsticks are on the left side of the ramen bowl - reconfirming our hypothesis.

In Table 2, we also observe improved performance using AIMv2 encoder and its 2D-RoPE version over other vision encoders in the LLaVA framework. AIMv2 design focuses on the image-first principle, where the model is trained to process all the image patches before decoding the text tokens in an autoregressive manner. This dense per-token supervision helps with tasks requiring fine-grained perception, such as in MMVP, CV-Bench, TallyQA, CountBenchQA, and VSR. We also believe that the two-stage captioning pipeline to generate AIMv2 training data helps in count-related examples, causing the LLaVA-AIMv2 variant to perform well in TallyQA and CountBenchQA compared to other spatial benchmarks (Lai et al., 2024).

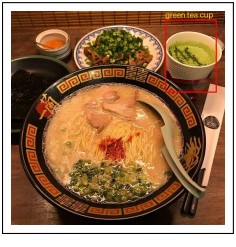 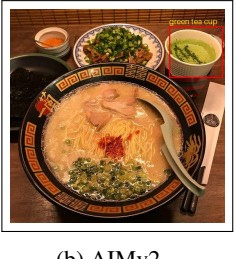 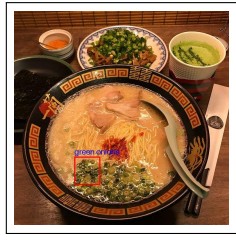 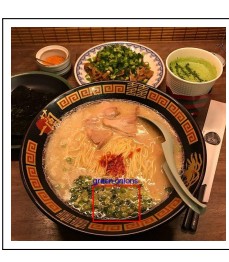

(a) SigLIP2     (b) AIMv2     (d) SigLIP2     (e) AIMv2

(c) Output for prompt: *Locate the cup that contains green liquid. Provide the bounding boxes.*

(f) Output for prompt: *Locate the pieces of green onions in the ramen bowl. Provide the bounding box coordinates given the size of the image is 550x550.*

Figure 4: Object localization in LLaVA-SigLIP2 vs. LLaVA-AIMv2 for different prompts.

Table 4 shows that Qwen2.5-VL achieves the strongest results among frontier models on all CV-Bench 2D tasks. Among our LLaVA variants, LLaVA-AIMv2 excels in CV-Bench 2D subtasks such as COCO and ADE20K. This is because the model's pre-training objective aligns with the nature of these tasks. COCO is for object detection and instance segmentation, and ADE20K is a scene parsing benchmark. Performance on these subtasks of CV-Bench 2D depends on the model's ability to capture fine-grained, pixel-level representations of spatial relationships. On the other hand, CLIP and SigLIP perform less effectively on these tasks. Their core objective is to learn a global vector for image-text matching, not a dense feature map. This limitation prevents them from achieving fine-grained visual understanding. Consequently, adding 2D-RoPE only helps to organize these global-level representations from these encoders and does not enable the model to learn any dense representations. SigLIP2's caption-based pretraining from LocCa (Wan et al., 2024) along with self-distillation and masked prediction from SILC (Naeem et al., 2023) and TIPS (Maninis et al., 2025)

explicitly directs the model to associate language to specific and relevant regions of the image. As a result, we see that SigLIP2 performs better than its predecessor in the subtasks of CV-Bench 2D.

An interesting observation in Table 4 is that LLaVA-SigLIP2 lags behind LLaVA-AIMv2 which might be unexpected given SigLIP2's masked image prediction objective. This is because the dense supervision is a secondary objective in SigLIP2, since dense supervision losses are added during the last 20% of the training regime(Tschannen et al., 2025). On the other hand, the core training objective of AIMv2 is based on learning the dense features from the very beginning. A qualitative example is shown in Figure 4 to demonstrate how LLaVA-AIMv2 is more precise in locating objects than LLaVA-SigLIP2.

As shown in Table 5, frontier model LLaVA-NeXT leads the GQA Query and Overall performance. PaliGemma leads GQA Choose and Logical. Among our trained LLaVA variants, the SigLIP-based models stand out: LLaVA-SigLIP-2D-RoPE achieves the highest scores on Overall, Choose, Compare, and Query, while LLaVA-SigLIP2-2D-RoPE leads on Logical and Verify. LLaVA-AIMv2-2D-RoPE shows moderate improvements over its baseline, particularly on Compare and Logical, but it does not surpass the SigLIP-based variants. We observe that Gemma3-4b-it surpasses the LLaVA variants, including the 2D-RoPE variants of SigLIP and SigLIP2 for GQA Choose subtask. We think this is because how Gemma3 is trained on supervised distillation from a large teacher model with a frozen visual encoder - making it an effective classifier. The Choose subtask of GQA is a constrained multi-task classification problem that aligns well with the training benefits of Gemma3. For other GQA subtasks, LLaVA 2D-RoPE variants of SigLIP, SigLIP2, and AIMv2 surpass Gemma3. We think that the structural advantage of 2D-RoPE performs well with the object and attribute relationships like GQA Compare, Logical. For substasks like GQA Query and Verify, the model needs to confirm information related to a specific object or attribute. SigLIP2's self-supervised losses and masked prediction result in rich local representations that are needed for these complex, compositional reasoning.

Table 4: Comparison of frontier models and LLaVA variants on CV-Bench 2D tasks. Values underlined indicate the best-performing model overall, while values in **bold** highlight the best-performing LLaVA encoder variant in each benchmark.

| Models | CV-Bench 2D COCO | CV-Bench 2D ADE20K | CV-Bench 2D Overall |
|---|---|---|---|
| LLaVA-NeXT | 0.680 | 0.532 | 0.606 |
| LLaVA-OneVision | 0.775 | 0.684 | 0.730 |
| Qwen2.5-VL | 0.820 | 0.689 | 0.754 |
| SmolVLM2 | 0.621 | 0.532 | 0.577 |
| Gemma3-4b-it | 0.660 | 0.618 | 0.659 |
| PaliGemma | 0.675 | 0.573 | 0.624 |
| Molmo | 0.773 | 0.684 | 0.728 |
| LLaVA-1.5 | 0.525 | 0.455 | 0.490 |
| LLaVA-2D-RoPE | 0.461 | 0.425 | 0.443 |
| LLaVA-SigLIP | 0.440 | 0.384 | 0.412 |
| LLaVA-SigLIP-2D-RoPE | 0.445 | 0.404 | 0.425 |
| LLaVA-SigLIP2 | 0.466 | 0.419 | 0.442 |
| LLaVA-SigLIP2-2D-RoPE | 0.436 | 0.393 | 0.415 |
| LLaVA-AIMv2 | **0.480** | **0.453** | **0.466** |
| LLaVA-AIMv2-2D-RoPE | 0.456 | 0.408 | 0.432 |

Table 5: Comparison of frontier models and LLaVA variants on GQA subtasks. Values underlined indicate the best-performing frontier model; values in **bold** indicate the best-performing LLaVA variant.

| Models | GQA Overall | GQA Choose | GQA Compare | GQA Logical | GQA Query | GQA Verify |
|---|---|---|---|---|---|---|
| LLaVA-NeXT | 63.786 | 85.120 | 64.177 | 78.980 | 49.875 | 82.860 |
| LLaVA-OneVision | 62.140 | 83.880 | 66.893 | 80.588 | 46.069 | 83.792 |
| Qwen2.5-VL | 60.391 | 84.322 | 73.854 | 78.702 | 43.027 | 82.682 |
| SmolVLM2 | 50.574 | 67.434 | 47.538 | 64.504 | 37.867 | 70.160 |
| Gemma3-4b-it | 31.277 | 61.913 | 16.978 | 19.301 | 25.952 | 45.337 |
| PaliGemma | 62.570 | 86.802 | 73.345 | 81.864 | 45.893 | 82.549 |
| Molmo | 55.295 | 78.742 | 52.462 | 65.169 | 41.558 | 77.886 |
| LLaVA-1.5 | 33.225 | 49.690 | 49.576 | 33.500 | 25.702 | 43.206 |
| LLaVA-2D-RoPE | 34.433 | 41.807 | 54.839 | 48.697 | 22.337 | 50.533 |
| LLaVA-SigLIP | 25.648 | 17.006 | 47.199 | 44.204 | 15.783 | 39.298 |
| LLaVA-SigLIP-2D-RoPE | **38.448** | **55.182** | **59.762** | 55.186 | **23.159** | 57.282 |
| LLaVA-SigLIP2 | 23.970 | 16.475 | 40.917 | 41.486 | 14.107 | 39.076 |
| LLaVA-SigLIP2-2D-RoPE | 34.560 | 47.121 | 55.688 | **57.959** | 15.298 | **62.211** |
| LLaVA-AIMv2 | 32.541 | 37.998 | 51.783 | 47.421 | 21.440 | 46.403 |
| LLaVA-AIMv2-2D-RoPE | 32.342 | 39.858 | 56.876 | 48.974 | 18.663 | 50.178 |

## 5 CONCLUSION

In our experiments, frontier models such as Qwen2.5-VL achieve the strongest overall results. Although Qwen2.5-VL is trained on a different dataset and token scales than LLaVA, the comparison is not strictly apples-to-apples. The more meaningful insights come from comparisons within the same VLM family, where architectural choices can be isolated. For example, CountBenchQA improves by about 58%, rising from 0.468 in LLaVA-1.5 to 0.739 in LLaVA-AIMv2, showing that encoder choice has a direct impact on spatial reasoning ability. The effects of injecting 2D-RoPE into the image encoder attention are mixed, indicating that where and how 2D positional information is introduced matter. Overall, the findings highlight that encoder design strongly shapes spatial awareness within VLM families, even if comparisons to frontier models remain questionable due to differences in training data and scale.

## 6 FUTURE WORK

Our study focused on static, 2D images, benchmarks and encoder variants within the LLaVA framework. This work can extend to 3D spatial reasoning along with the dynamic environment. Another potential extension can be on SigLIP2 with NaFlex. The flexible resolution image preprocessing of NaFlex mitigates information loss observed in fixed-resolution encoders. In terms of visual backbones, incorporating DINOv2 in LLaVA is left out for future work. Similarly BLIP-2 and related architectures could help assess whether their pretraining objectives and visual-language alignment strategies offer advantages over CLIP-derived models. Finally, we note that advanced alignment mechanisms such as gated attention in Flamingo Alayrac et al. (2022), Q-Former in BLIP-2, or cross-modal pooling in MM1 McKinzie et al. (2024) were intentionally set aside in this work. Exploring these approaches as alternatives to simple projection layers may further improve spatial reasoning performance and cross-modal integration.

## 7 REPRODUCIBILITY STATEMENT

The code and dataset for this paper are open source and will be released upon acceptance. For reproduction purposes, please refer to Section 3.

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

# A   APPENDIX

In Table 6, LLaVA-SigLIP-2D-RoPE emerges as the strongest LLaVA variant overall, leading on 3D Overall and Distance, while LLaVA-SigLIP has the best Depth score. Specifically on Depth, LLaVA-AIMv2-2D-RoPE improves about 2.7% over its baseline. Among frontier models, LLaVA-OneVision dominates across 3D metrics, with Qwen2.5-VL excelling on 3D Distance.

Table 6: Comparison of frontier models and LLaVA variants on CV-Bench 3D tasks (Overall, Depth, Distance). Values underlined indicate the best-performing frontier model; values in **bold** indicate the best-performing LLaVA variant.

| Models | CV-Bench 3D Overall | CV-Bench 3D Depth | CV-Bench 3D Distance |
|---|---|---|---|
| LLaVA-NeXT | 0.627 | 0.693 | 0.560 |
| LLaVA-OneVision | 0.769 | 0.820 | 0.718 |
| Qwen2.5-VL | 0.736 | 0.698 | 0.773 |
| SmolVLM2 | 0.475 | 0.458 | 0.492 |
| Gemma3-4b-it | 0.557 | 0.463 | 0.520 |
| PaliGemma | 0.441 | 0.442 | 0.440 |
| Molmo | 0.729 | 0.798 | 0.660 |
| LLaVA-1.5 | 0.642 | 0.717 | 0.567 |
| LLaVA-2D-RoPE | 0.614 | 0.725 | 0.503 |
| LLaVA-SigLIP | 0.376 | **0.730** | 0.022 |
| LLaVA-SigLIP-2D-RoPE | **0.649** | 0.703 | **0.595** |
| LLaVA-SigLIP2 | 0.375 | 0.698 | 0.052 |
| LLaVA-SigLIP2-2D-RoPE | 0.503 | 0.712 | 0.293 |
| LLaVA-AIMv2 | 0.361 | 0.703 | 0.018 |
| LLaVA-AIMv2-2D-RoPE | 0.387 | 0.722 | 0.052 |

As shown in Tables 7, 8, and 9 with respect to Figure 3, we observe that the model outputs frequently contain hallucinations, highlighted in red. We see two different types of hallucinations here: (i) the models sometimes hallucinates on objects that do not exist in the image, such as a napkin, vase, or fork; and (ii) when existing objects are mentioned, their spatial placement is often misidentified, such as utensils resting in the wrong position on the table. These qualitative results illustrate how current models struggle with grounding object presence and location, even though the containing a coherent description.

Table 7: Qwen output for the prompt *describe this image* in reference to Figure 3.

| Model | Output |
|---|---|
| Qwen2.5-VL | The image shows a meal consisting of a bowl of ramen, a side dish, and a cup of green tea. The ramen bowl is filled with a light-colored broth, topped with noodles, slices of pork, green onions, and a sprinkle of red chili flakes. The bowl has a decorative black and gold rim. To the side, there is a plate with what appears to be a serving of grilled or cooked meat garnished with green onions. Additionally, there is a small bowl of orange-colored seasoning, likely for the ramen, and a cup of green tea with a lid, which is partially visible. The table setting includes chopsticks and a napkin, suggesting a traditional dining experience. The background features a bamboo mat, adding to the ambiance of the setting. |

Table 8: LLaVA with image encoders for the prompt *describe this image* in reference to Figure 3.

| Model | Output |
|---|---|
| LLaVA-1.5 | The image features a dining table with a variety of food items, including a large bowl of soup, a plate of noodles, and a bowl of vegetables. The soup is placed in the center of the table, while the noodles and vegetables are positioned around it. In addition to the main dishes, there are several bowls and cups scattered across the table, possibly containing other food items or beverages. A spoon is also visible on the table, likely used for eating the soup. The table setting creates an inviting atmosphere for a meal shared among friends or family. |
| LLaVA-SigLIP | The image features a dining table with a large bowl of soup placed in the center. The soup is filled with various ingredients, including noodles, meat, and vegetables. The bowl is accompanied by a spoon, which is resting on the table. In addition to the main bowl, there are several other bowls and cups scattered around the table. Some bowls are placed near the edge of the table, while others are located closer to the center. A cup can be seen on the left side of the table, and another cup is positioned towards the right side. The table is also adorned with a few decorative elements, such as a vase and a potted plant. |
| LLaVA-SigLIP2 | The image features a dining table with a variety of food items and utensils. There is a bowl of soup placed in the center of the table, accompanied by a spoon and a fork. Another bowl is located towards the left side of the table, and a cup can be seen on the right side. In addition to the main bowl of soup, there are two smaller bowls, one on the left side and another on the right side of the table. A cup is also present on the right side of the table. The table is set with a variety of food items, including a sandwich and a carrot. |
| LLaVA-AIMv2 | The image features a dining table with a variety of food items and utensils. There is a bowl of noodles with meat and vegetables, including broccoli, placed in the center of the table. A spoon is resting in the bowl, ready to be used for eating. In addition to the main bowl, there are several other bowls and cups scattered around the table, containing different food items. A cup of soup is placed near the top right corner of the table, while another bowl is located towards the left side. A couple of chopsticks are present on the table, and a fork is also visible. The table is set for a meal, with a variety of dishes and utensils. |

Table 9: LLaVA variants with 2D-RoPE for the prompt *describe this image* in reference to Figure 3.

| Model | Output |
|---|---|
| LLaVA-2D-RoPE | The image features a dining table with a large bowl of soup placed in the center, surrounded by various bowls and plates of food. The soup appears to be a noodle dish, and there are several pieces of broccoli scattered around the table. In addition to the main bowl, there are three smaller bowls placed on the table. A spoon is also visible near the center of the scene. The table is set with a variety of food items, making it an inviting and appetizing scene. |
| LLaVA-SigLIP-2D-RoPE | The image features a delicious meal served in a bowl, which includes noodles, meat, and vegetables. The bowl is placed on a dining table, and there are several pieces of broccoli scattered around the bowl. In addition to the main dish, there are two cups on the table, one near the top left corner and the other towards the top right corner. A spoon is also present, resting on the table near the bowl. The overall setting appears to be a cozy and inviting meal. |
| LLaVA-SigLIP2-2D-RoPE | The image features a dining table with a variety of food and drinks. There is a bowl of soup placed in the center of the table, accompanied by a spoon. A cup of tea is also present, along with a cup of coffee. In addition to the main dishes, there are several other bowls and cups scattered around the table, possibly containing different types of food or beverages. A fork can be seen on the table, and a bottle is located near the edge. |
| LLaVA-AIMv2-2D-RoPE | The image features a dining table with a large bowl of soup placed in the center, surrounded by various bowls and cups. The soup appears to be a noodle soup, and there are several spoons placed around the table, ready for use. In addition to the soup, there are two bowls of food and a sandwich on the left side. A couple of cups can also be seen. The overall scene suggests a meal being enjoyed by multiple people. |

## B  USAGE OF LARGE LANGUAGE MODELS

Large Language Models (LLMs) were only used to polish text, diagrams, and create a LaTeX table format.

