# OpenReview forum: "The Spatial Blindspot of Vision-Language Models"
_ICLR.cc/2026/Conference — ICLR 2026 Conference Withdrawn Submission_

### Official Review · Reviewer_JmNX · 2025-10-27

**Soundness:** 2
**Presentation:** 2
**Contribution:** 1
**Rating:** 0
**Confidence:** 5

**Summary:**

This paper investigates the long-standing issue of spatial reasoning deficiencies in vision-language models (VLMs). Built on the LLaVA framework, it explores two main directions to enhance spatial awareness: (1) integrating alternative image encoders such as SigLIP, SigLIP2, and AIMv2 instead of CLIP, and (2) injecting explicit two-dimensional rotary position embeddings (2D-RoPE) to preserve spatial layout in the image–language alignment. The authors benchmark these variants across several spatial reasoning datasets (e.g., CV-Bench, VSR, GQA) and report modest improvements in some cases. While the paper frames spatial awareness as a critical missing capability in current VLMs, its contributions, adding 2D-RoPE and swapping visual backbones, largely replicate techniques already established in prior work, such as Qwen2-VL (which introduced M-RoPE (kind of 2D RoPE)) and Cambrian-1 (which systematically studied the effect of SigLIP-based vision encoders). Consequently, the work provides incremental empirical observations rather than a novel conceptual or methodological advance.

**Strengths:**

The paper presents a strong and well-motivated problem statement, and the authors provide extensive experiments and ablations demonstrating that the choice of vision encoder and the design of positional encoding play an important role in improving spatial reasoning in vision-language models.

**Weaknesses:**

The main weakness of the paper lies in its limited novelty. Both proposed directions, introducing 2D rotary positional embeddings and replacing the CLIP encoder with stronger vision backbones such as SigLIP or AIMv2, have already been explored in prior work. Specifically, Qwen2-VL has introduced M-RoPE for preserving spatial structure, and Cambrian-1 has systematically studied the effect of different vision encoders, including SigLIP, on multimodal reasoning. As a result, the paper’s contributions are largely incremental, offering empirical confirmation rather than conceptual or methodological innovation.

**Questions:**

Could the authors clarify what they consider the novel contribution of this work, given that both 2D-RoPE and the analysis of alternative vision encoders (e.g., SigLIP, AIMv2) have already been explored in prior models such as Qwen2-VL and Cambrian-1? If the authors believe their approach differs, please justify this distinction clearly and explain what new insight or methodological advancement this paper provides beyond those existing studies.

---

### Official Review · Reviewer_k2vx · 2025-10-29

**Soundness:** 3
**Presentation:** 3
**Contribution:** 2
**Rating:** 6
**Confidence:** 3

**Summary:**

This paper takes aim at a core limitation of modern Transformer-based vision encoders — the spatial flattening of input images into patches and the use of 1D positional encodings. The authors argue that this limits their spatial reasoning abilities and aim to ameliorate this core issue using a variety of visual encoders  with and without 2-dimensional positional encodings.
The general takeaway is that while there are some effects of changing the encoder on spatial reasoning abilities, the specific effects are more mixed and certain frontier models like QwenVL continue to outperform the fine-tuned models the authors produce and other frontier models perhaps hinting at the importance of other architectural and training data decisions.

**Strengths:**

* I commend the authors on the range of encoder architectures evaluated and the decision to include variants with and without the 2D positional encoding
* The paper is generally well written and easy to follow, with clear motivations and thoughtful error analyses. I found the targeted comparisons between specific model pairs and discussion surrounding why the results might be arise helpful and a point in favor of the paper.
* I think the introduction provides a particularly strong overview of the current state of VLM systems. While not directly related to the experiments at hand, I found this background helpful especially for readers who might not be as involved in the computer vision community.

**Weaknesses:**

* While I overall enjoyed reading the paper, I think the mixed nature of the results makes it difficult for me to synthesize a key takeaway. I particularly don’t feel like this sentence from the Conclusion is justified `Overall, the findings highlight that encoder design strongly shapes spatial awareness within VLM families`. While there were differences in overall performance between models, the difference among the fine-tuned models were quite muted (with some notable exceptions that the authors note). In fact, given that all 3 visual encoders get at least one ‘win’ and sometimes even without the need for 2D positional encodings, it is unclear how the next generation of visual encoders for spatial reasoning should look.

* While I also agree that it is difficult to compare across different model families, the fact that performance of the fine-tuned models falls far below the other frontier models at times (especially when some models don’t use 2D encodings) makes me think whether these differences between models would disappear at larger model scales or with more training tokens.

* While, as I note, I did find the individual error analyses helpful, I think the authors draw overly strong conclusions from select key examples at times. For example, I don’t find the differences between SigLIP2 and AIMv2 starkly different in Figure 4. Similarly, the finding surrounding Gemma3-4b mislabeling the position of chopsticks also feels somewhat stochastic? Definitely not something that would merit the term `reconfirming our hypothesis`. I would recommend the authors tamper their language about the results.

**Questions:**

* I have no particular questions beyond the authors needing to potentially address some of the points raised in the ‘weaknesses’ section. I would say there is a need for a clear conclusion statement about the findings here that is backed up by empirical results. In the absence of clear results, the authors should acknowledge that more work needs to be done to further isolate ways to improve spatial reasoning in VLMs.

* Fix citing formatting. Often author names are repeated e.g., `Zhang et al. (Zhang et al., 2025a)`. Use the appropriate `\citeauthor` command in these cases.

* I think the authors need to include an explicit limitations section.

---

### Official Review · Reviewer_gb8H · 2025-10-31

**Soundness:** 2
**Presentation:** 1
**Contribution:** 3
**Rating:** 4
**Confidence:** 5

**Summary:**

This paper addresses the "spatial blindspot" in Vision-Language Models (VLMs), attributing it to two dominant architectural choices: (1) reliance on contrastive encoders like CLIP and (2) flattening image patches into 1D sequences, which discards 2D structure. To investigate this, the authors conduct a study within the LLaVA framework by testing alternative encoders (SigLIP, SigLIP2, AIMv2) and 2D Rotary Position Embeddings (2D-ROPE). The study finds that encoder choice significantly impacts spatial abilities (e.g., AIMv2 improves counting). However, while 2D-ROPE was hypothesized to enhance spatial understanding, the experimental results show its effects were mixed and, in several key instances, negative.

**Strengths:**

1. Important Problem: The paper tackles a critical and widely acknowledged weakness in VLMs—spatial reasoning—which is a known bottleneck for applications like Embodied AI.

2. Systematic Design: The core design of the study—comparing different encoder objectives and positional encodings within the controlled LLaVA framework—is systematic and a sound scientific approach.

3. Comprehensive Benchmarking: The authors evaluate performance on a large (7+) and diverse suite of specialized spatial benchmarks (e.g., MMVP, CV-Bench, VSR, TallyQA), providing a nuanced view of model capabilities.

4. Clear Empirical Finding (re: Encoders): The paper's strongest and most valuable contribution is its clear empirical finding that the image encoder's pre-training objective has a direct and significant impact on spatial reasoning. The 58% improvement on CountBenchQA from LLaVA-1.5 to LLaVA-AIMv2 is a clear demonstration of this.

**Weaknesses:**

1. Scattered Logic and Misaligned Experimental Focus: The paper's core scientific contribution should be the intra-LLaVA ablation study. However, the experimental analysis is completely unfocused. The narrative erratically jumps between this controlled study and extensive "SOTA-comparisons" against frontier models like Qwen2.5-VL. The authors later dismiss these comparisons as not "apples-to-apples," which makes their inclusion distracting and confusing. This scattered focus makes the paper's core argument difficult to follow.

2. Flaws in argument structure: the paper’s organization is fundamentally flawed, undermining its soundness. The introduction overly focuses on listing points with little logical coherence between paragraphs, and the experimental analysis lacks a clear central focus. key quantitative evidence (such as the “58% improvement” on CountBenchQA) is inappropriately withheld until the conclusion rather than analyzed in the previous section, severely violating scientific writing norms.

3. Failed Hypothesis and Missing Analysis (2D-ROPE): A central hypothesis and claimed contribution of the paper is that 2D-ROPE will "enhance" spatial reasoning. The data clearly shows this hypothesis failed. On several key spatial benchmarks (e.g., CV-Bench 2D, TallyQA, CountBenchQA), the 2D-ROPE variant performed worse than the 1D baseline. Critically, the paper provides zero analysis or discussion as to why adding explicit 2D information would harm spatial performance. This is a major scientific omission.

4. Missing Key Experiments: As a direct result of the misaligned focus (e.g., on Qwen), critical experiments were not performed. Most notably, a DINOv2 baseline is missing, which is essential for validating the "dense feature" hypothesis. Furthermore, the paper "intentionally set aside" alignment mechanisms (like Q-Former), which is a major confounding variable.

5. Poor Clarity and Errors: The paper is difficult to read, saturated with technical jargon. Figures (like Figure 2) are overly abstract and uninformative. There are also clear errors in the data presentation, such as the incorrect highlighting of the best-performing model in Table 1 for the MME metric.

**Questions:**

1. On Experimental Organization: The results are saturated with comparisons to frontier models like Qwen2.5-VL, which the conclusion then dismisses as not "apples-to-apples." What is the scientific justification for including these extensive, non-controlled, and distracting comparisons that derail the paper's core ablation study?

2. On 2D-ROPE's Negative Results: One of the paper's core hypotheses appears to have failed, with 2D-ROPE degrading performance on key spatial benchmarks for the best-performing AIMv2 variant. The paper provides no analysis for this. Can the authors provide a hypothesis for this phenomenon?

3. On Missing Key Experiments: The paper's focus on dense features is missing the most critical DINOv2 baseline. Would the authors be willing to provide this? Furthermore, since alignment was "intentionally set aside," how can the authors be sure the bottleneck is the encoder and not LLaVA's simple linear projection?

---

### Official Review · Reviewer_tS3F · 2025-10-31

**Soundness:** 2
**Presentation:** 2
**Contribution:** 2
**Rating:** 4
**Confidence:** 5

**Summary:**

This paper investigates why current VLMs struggle with spatial reasoning and attributes the issue to architectural choices rather than data or fine-tuning. The authors argue that contrastive image encoders like CLIP learn globally semantic but spatially coarse features, and that 1D rotary positional embeddings (RoPE) further destroy 2D structure.

Using the LLaVA framework, they systematically vary visual encoders (CLIP, SigLIP, SigLIP2, AIMv2) and positional encodings (1D vs. 2D-RoPE) under identical training conditions. Across benchmarks such as CV-Bench, GQA, and VSR, densely or autoregressively pretrained encoders (SigLIP2, AIMv2) outperform CLIP by about 7–10 points on spatial tasks, while 2D-RoPE yields only minor gains.

The study concludes that the main cause of spatial blindness lies in encoder pretraining objectives, not fine-tuning or data scale. It provides a useful diagnostic of where spatial reasoning fails but offers no concrete architectural or algorithmic solution.

**Strengths:**

1. The paper clearly isolates architectural sources of spatial reasoning failure in VLMs, shifting attention from data or fine-tuning toward the underlying design of image encoders and positional embeddings.

2. By varying only the encoder and positional encoding within an identical LLaVA pipeline, the study provides a controlled and fair comparison that strengthens the validity of its empirical observations.

3. The evaluation spans multiple spatial reasoning and general multimodal benchmarks, offering a broad and consistent empirical picture of how encoder pre-training objectives affect spatial performance.

4. The finding that dense or autoregressive supervision (e.g., SigLIP2, AIMv2) yields better spatial awareness provides a concrete design recommendation for improving future visual encoders.

**Weaknesses:**

1. The paper identifies the causes of spatial blindness but does not propose or evaluate a real fix beyond minor encoder swaps and a simple 2D-RoPE variant, limiting its contribution to diagnosis rather than advancement.

2. The conclusion that dense or autoregressive pre-training improves spatial reasoning is based solely on benchmark trends, without any feature-level analysis, visualisation, or causal verification.

3. The encoders compared differ not only in pre-training objectives but also slightly in scale and training strength, making it unclear whether the observed gains stem from supervision type or overall model capacity.

4. The positional encoding modification yields only marginal and inconsistent improvements, suggesting limited practical relevance and insufficient justification for its inclusion as a technical contribution.

5. The spatial limitations of CLIP-style encoders are well-documented in recent works such as SpatialVLM and SpatialRGPT, making the problem setup familiar and the novelty of the findings relatively low.

**Questions:**

1. Since the paper primarily diagnoses the causes of spatial blindness without proposing a tangible fix, what specific new insight or mechanism does it contribute beyond confirming previously reported limitations of CLIP-style encoders?

2. The results suggest that dense or autoregressive encoders perform better, but have the authors conducted any analysis to verify that spatial gains arise from supervision type rather than other correlated factors?

3. Given that the compared encoders differ slightly in capacity and training budgets, how do the authors disentangle improvements due to architectural or objective differences from those caused by scale?

4. The proposed positional encoding shows marginal and inconsistent benefits. Can the authors justify its inclusion as a meaningful technical contribution or explain under what conditions it helps?

5. Have the authors analyzed attention maps, feature activations, or other internal representations to provide evidence that denser pre-training leads to more spatially grounded features?

---

### Official Review · Reviewer_gU3V · 2025-11-03

**Soundness:** 2
**Presentation:** 2
**Contribution:** 2
**Rating:** 2
**Confidence:** 4

**Summary:**

The paper investigates spatial reasoning limitations of Vision-Language Models (VLMs), focusing on two aspects within the LLaVA framework:

1.	Using alternative image encoders (AIMv2, SigLIP, SigLIP2).
2.	Applying 2D rotary positional embeddings (2D-RoPE).

The authors evaluate these on several benchmarks (MMVP, GQA, TallyQA, CountBenchQA, etc.) and claim modest gains in spatial tasks.

**Strengths:**

- The paper focuses on an important and often underexplored aspect of Vision-Language Models (VLMs): spatial awareness. Improving spatial reasoning is crucial for grounding and physical understanding in multimodal models.

- The authors attempt to address this by introducing 2D rotary positional embeddings within the LLaVA framework and by evaluating few alternative image encoders (e.g., SigLIP, SigLIP2, AIMv2) to study their impact on spatial reasoning performance.

**Weaknesses:**

1. **Overstated Novelty: Prior Work Already Explores 2D or Multi-Dimensional Positional Embeddings**

Prior work has already integrated 2D (or multi-dimensional) positional embeddings in vision or vision-language models, so the paper’s claim that this area is “under-explored” is inaccurate.

Qwen2-VL introduced Multimodal Rotary Positional Embedding (M-RoPE), which decomposes positional embeddings into three parts capturing 1D text, 2D visual (height/width), and 3D video temporal information [1].

Rotary Position Embedding for Vision Transformer studied 2D RoPE in Vision Transformers and showed that extending RoPE to 2D vision improves spatial representation and performance [2].

Because of this, the paper’s novelty claim about 2D embeddings for spatial reasoning in VLMs is weakened.

2. **Lack of New Insight or Strong Contribution**

The core idea, adding 2D positional embeddings and testing different image encoders to improve spatial reasoning, is relatively obvious. Giving a model explicit spatial cues is an expected way to enhance spatial awareness.

The paper does not propose a new component (algorithm, architecture, or loss) specifically designed for spatial reasoning beyond combining already existing encodings and backbones.

There is minimal analysis exploring why or how much the 2D positional embedding improves reasoning, such as through attention map visualization or failure mode analysis.

3. **Empirical Gains Are Inconsistent and Limited**

When adding 2D-RoPE, some benchmarks such as MME show a drop of about 1.75% instead of consistent improvement.

Performance gains across spatial reasoning benchmarks are small and do not clearly surpass prior models.

4. **Weak Experimental Design and Controls**

Comparisons between encoder variants are confounded by differences in backbone scale, pretraining datasets, and fine-tuning setups, making it difficult to isolate the true effect of positional encoding versus encoder choice.

The spatial reasoning task definition is broad, but the paper lacks rigorous probing of spatial structure (e.g., distinguishing left/right or near/far relations).

**References**

[1] Wang, P., et al. Qwen2-VL: Enhancing Vision-Language Model's Perception of the World at Any Resolution. arXiv preprint arXiv:2409.12191, 2024.

[2] Heo, B., et al. Rotary Position Embedding for Vision Transformer. arXiv preprint arXiv:2403.13298, 2024.

**Questions:**

### Positioning vs. prior work:
- How does this work differ technically from Qwen2-VL [1], which already employs a 2D (and 3D) rotary positional embedding (M-RoPE)?
- Can the authors explain what unique insight your paper provides beyond confirming that 2D positional encodings help spatial reasoning?

### Definition and Diagnosis of “Spatial Blindspot”
- The authors frame spatial blindness as a key limitation, yet it is not explicitly quantified. How did you determine that failures on spatial benchmarks arise specifically from positional encoding issues and not from limited pretraining or instruction data?
- Could the “blindspot” be more a data bias problem than an architectural one? Have the authors tested this by fine-tuning with spatially dense supervision (e.g., region captions, referring expressions)?

### Encoder Comparisons
- The models compared (CLIP, SigLIP, AIMv2) differ in pretraining objectives, data scale, and architecture. How do the authors ensure that differences in performance are attributable to spatial reasoning and not dataset bias or model capacity?
- AIMv2 shows stronger results on spatial tasks. Did the authors analyze whether this is due to 2D positional structure, more localized attention, or richer pretraining signals (e.g., bounding box supervision)?

---

### Note · Authors · 2025-11-14

I have read and agree with the venue's withdrawal policy on behalf of myself and my co-authors.